# Benign and Malignant Outcomes in the Offspring of Females Exposed In Utero to Diethylstilbestrol (DES): An Update from the NCI Third Generation Study

**DOI:** 10.3390/cancers16142575

**Published:** 2024-07-18

**Authors:** Linda Titus, Elizabeth E. Hatch, Kimberly A. Bertrand, Julie R. Palmer, William C. Strohsnitter, Dezheng Huo, Michael Curry, Marianne Hyer, Kjersti Aagaard, Gretchen L. Gierach, Rebecca Troisi

**Affiliations:** 1Department of Pediatrics, Geisel School of Medicine at Dartmouth, and the Norris Cotton Cancer Center, Lebanon, NH 03756, USA; lindatitusdartmouth@gmail.com; 2Department of Epidemiology and Biostatistics, Boston University School of Public Health, Boston, MA 02118, USA; eehatch@bu.edu; 3Boston University Chobanian & Avedisian School of Medicine and Slone Epidemiology Center, Boston University, Boston, MA 02118, USAjpalmer@bu.edu (J.R.P.); 4Department of Molecular, Cell and Cancer Biology, University of Massachusetts Medical School, Worcester, MA 01655, USA; william.strohsnitter@umassmed.edu; 5Department of Public Health Sciences, University of Chicago, Chicago, IL 60637, USA; 6Information Management Services, Rockville, MD 20852, USA; 7Department of Obstetrics & Gynecology, Division of Maternal-Fetal Medicine, Baylor College of Medicine and Texas Children’s Hospital, Houston, TX 77401, USA; 8Division of Cancer Epidemiology and Genetics, National Cancer Institute, Bethesda, MD 20892, USA; gierachg@mail.nih.gov

**Keywords:** diethylstilbestrol, in utero, prenatal exposure, cancer, third generation, endocrine disruptor

## Abstract

**Simple Summary:**

Females prenatally exposed to diethylstilbestrol have an elevated risk of severe cervical dysplasia and some cancers, while testicular cancer is increased in males. We assessed these associations in the prenatally exposed female’s offspring (third generation). Based on third-generation females’ self-reports, diethylstilbestrol exposure was not associated with risks of overall cancer, breast cancer, or severe cervical dysplasia. Borderline ovarian cancer risk in exposed vs. unexposed was elevated but compatible with chance. Based on mothers’ reports, diethylstilbestrol exposure did not increase the risk of overall or other cancers in third-generation females. Overall cancer risk in exposed males appeared elevated but compatible with chance. Testicular cancer risk was not elevated in exposed males, and there were no prostate cancers reported. These data do not provide evidence that diethylstilbestrol is associated with cancer risk in third-generation females or males. The third generation is relatively young and requires follow-up to assess risk as the cohort ages.

**Abstract:**

Background: Females exposed prenatally to diethylstilbestrol (DES) have an elevated risk of cervical dysplasia, breast cancer, and clear cell adenocarcinoma (CCA) of the cervix/vagina. Testicular cancer risk is increased in prenatally exposed males. Epigenetic changes may mediate the transmission of DES effects to the next (“third”) generation of offspring. Methods: Using data self-reported by third-generation females, we assessed DES in relation to the risk of cancer and benign breast and reproductive tract conditions. Using data from prenatally DES-exposed and unexposed mothers, we assessed DES in relation to cancer risk in their female and male offspring. Cancer risk was assessed by standardized incidence ratios (SIR) and 95% confidence intervals (CI); the risks of benign and malignant diagnoses were assessed by hazard ratios (HR) and 95% CI. Results: In self-reported data, DES exposure was not associated with an increased risk of overall cancer (HR 0.83; CI 0.36–1.90), breast cancer, or severe cervical dysplasia. No females reported CCA. The risk of borderline ovarian cancer appeared elevated, but the HR was imprecise (3.46; CI 0.37–32.42). Based on mothers’ reports, DES exposure did not increase the risk of overall cancer (HR 0.80; CI 0.49–1.32) or of other cancers in third-generation females. Overall cancer risk in exposed males appeared elevated (HR 1.41; CI 0.70–2.86), but the CI was wide. The risk of testicular cancer was not elevated in exposed males; no cases of prostate cancer were reported. Conclusions: To date, there is little evidence that DES is associated with cancer risk in third-generation females or males, but these individuals are relatively young, and further follow-up is needed.

## 1. Introduction

Starting in the early 1940s, diethylstilbestrol (DES), a synthetic non-steroidal estrogen, was prescribed to pregnant females to reduce the risk of pregnancy complications and losses. In the early 1950s, clinical trials showed no benefit from DES during pregnancy [1,2]. Nevertheless, DES continued to be prescribed to pregnant females until 1971, when the US FDA published a warning [3] based on a study showing a strong link between prenatal exposure and vaginal/cervical clear cell adenocarcinoma (CCA) in young females [4]. Subsequent studies of health outcomes in prenatally DES-exposed females confirmed the DES-CCA association [5,6] and found increased risks of higher-grade cervical intraepithelial neoplasia (CIN2+) [7,8] and breast cancer in older women [9,10]. In addition, studies of prenatally exposed males [11], including a meta-analysis [12], indicated an increased risk of testicular cancer.

DES is an established transplacental carcinogen in humans. Although no longer given in pregnancy, it stands as a model of potential intergenerational chemical toxicity and is particularly relevant to the health effects of exposure to other endocrine-disrupting chemicals. The impact of this drug on prenatally exposed offspring, particularly females, is clear [10]. A current question is whether DES also affects the third generation, i.e., the offspring of prenatally exposed individuals. The possibility that adverse effects are transmitted to the third generation is suggested by studies showing an increased frequency of tumors in the offspring of prenatally DES-exposed mice [13,14,15]. To date, an influence of DES on germline DNA has not been identified. However, evidence from mouse [16] and human studies [17] suggests that epigenetic changes, including DNA methylation, may mediate the intergenerational effects of DES. More broadly, recent evidence from animal and human studies supports an intergenerational effect of exposure to certain endocrine disrupting chemicals, and emerging evidence suggests that epigenetic changes to the germ cells of exposed parents may impact the health of offspring [18,19].

A preliminary report from the US National Cancer Institute (NCI) Third Generation Study, a component of the NCI multi-generation DES Follow-up Study, did not support an association between DES and overall cancer risk but suggested an increase in borderline ovarian tumors in young third-generation females [20].

In the current report, which reflects nearly 20 years of additional data collection, we update associations between DES and the risk of cancer and benign conditions affecting the breast and reproductive tract in females participating in the NCI Third Generation Study. We expanded the study further by adding updated data from prenatally DES-exposed and unexposed females (mothers) to identify cancers in third-generation male offspring and in females who were not participating in the Third Generation Study.

## 2. Methods

### 2.1. Cohorts and Data Collection

The NCI DES Follow-up Study, initiated in 1992, originally comprised two generations: females exposed and unexposed to DES during pregnancy (first generation females) and their prenatally exposed and unexposed offspring (second-generation females and males) [5,11]. Questionnaire mailings to second-generation study participants took place in 1994, 1997, 2001, 2006, 2011, and 2016. Each phase of second-generation data collection queried females for cancer diagnoses affecting their female and male offspring.

In 2000, the NCI expanded the DES study to include the daughters (third-generation females) of prenatally exposed and unexposed females participating in the DES Follow-up Study. The methods of the Third Generation Study have been described previously [21]. Briefly, second-generation females (prenatally exposed and unexposed) whose parity records indicated female offspring were asked to provide contact information for daughters who were at least 18 years of age. Third-generation females for whom contact information was received were eligible for entry into the cohort registry and were approached for study enrollment. The collection of baseline data took place in three phases as females reached 18 years of age: 2001, 2009, and 2019. Females who previously participated in baseline data collection were approached for follow-up in 2009 and 2019. The baseline and follow-up questionnaires were completed by mail, telephone, or online using a secure, web-based instrument.

The initial review of parity records identified 1781 (966 exposed and 815 unexposed) third-generation females who were age 18 or more. Second-generation mothers provided contact information for 516 (53.4%) exposed and 382 (46.9%) unexposed third-generation females; of these, 464 (89.9%) exposed and 329 (86.1%) unexposed females returned a baseline questionnaire. The parity review for the 2009 baseline enrollment period identified 1195 (857 exposed, 338 unexposed) newly age-eligible third-generation females. Mothers provided contact information for 416 (48.5%) exposed and 161 (47.6%) unexposed third-generation females; of these, 334 (80.0%) exposed and 138 (86.3%) unexposed females returned a baseline questionnaire. One of the five study centers (Texas) was unable to participate in the 2019 baseline and follow-up data collection. The parity review for the 2019 baseline enrollment identified 649 (459 exposed and 190 unexposed) newly age-eligible third-generation females at the four contributing study centers. Mothers provided contact information for 236 (51.4%) exposed and 75 (39.5%) unexposed third-generation females; of these, 166 (70.3%) exposed and 62 (82.7%) unexposed returned a baseline questionnaire.

In 2009, we sent follow-up questionnaires to 789 (462 exposed, 327 unexposed) third-generation females enrolled in the 2001 baseline study (4 could not be re-approached due to Institutional Review Board regulations at one study center). Of these, 381 (82.5%) exposed and 280 (85.6%) unexposed females returned a questionnaire. In 2019, the four contributing study centers sent follow-up questionnaires to 638 exposed and 433 unexposed third-generation females. Of these, 763 females, 458 (71.8%) exposed and 305 (70.4%) unexposed, participated in follow-up. In all, baseline questionnaires were available for 964 exposed and 529 unexposed third-generation female participants. At least one follow-up questionnaire was completed by 592 (80.7%) of the exposed and 390 (85.7%) of the unexposed (the denominators used to calculate these percentages take into account that one of the study centers did not participate in the 2019 follow-up). The average duration of follow-up over the entire study period was 13.3 years for the exposed and 14.1 years for the unexposed.

### 2.2. Covariates

All Third Generation Study questionnaires queried participants for demographic characteristics and health-screening practices.

### 2.3. Cancer Outcomes

Questions regarding cancer diagnosis, type of cancer, and year of diagnosis were included on all questionnaires administered to Third Generation Study participants. Similar questions regarding cancer diagnoses affecting female and male offspring were included on all questionnaires administered to second-generation females (mothers) participating in the NCI DES Follow-up Study.

### 2.4. Nonmalignant Outcomes

Third Generation Study participants were asked to report benign diagnoses of the breast and reproductive tract. Self-reported nonmalignant breast biopsy outcome groups were atypia (e.g., atypical ductal hyperplasia, atypical lobular hyperplasia, breast carcinoma in situ); benign breast tumor (e.g., fibroadenoma, lipoma); cyst (e.g., cyst, fibrocystic disease); and unspecified diagnoses. Self-reported ovarian outcome groups were borderline ovarian tumors (tumors of low malignant potential); benign tumors; cysts; and polycystic ovarian syndrome (PCOS) (diagnosis of PCOS was available only from the 2019 follow-up questionnaire). Diagnoses of moderate/severe dysplasia of the lower genital tract (referred to here as CIN2+) were identified through pathology reports retrieved for Third Generation Study participants who reported a lower genital tract biopsy. Moderate/severe dysplasia included CIN2, CIN3, high-grade squamous intraepithelial lesion (HGSIL), and carcinoma in situ (CIS). Pathology reports were available for 112/239 (46.9%) of the exposed and 75/138 (54.3%) of the unexposed females who reported a lower genital tract biopsy. Of the 112 exposed women with pathology results, 29 (26%) had a diagnosis of CIN2+. Of the 75 unexposed women with pathology results, 21 (28%) had a diagnosis of CIN2+. When multiple diagnoses affected the same female, only the most severe diagnosis was assessed.

### 2.5. Statistical Analysis

We conducted separate analyses for benign diagnoses and for cancers self-reported by Third Generation Study participants, for mothers’ reports of cancers affecting third-generation females, and for mothers’ reports of cancers affecting third-generation males. An additional analysis combined cancers in sons and daughters as reported by mothers.

Analyses were conducted using SAS software (Version 9.4) [22]. Standardized incidence ratios (SIR) with exact 95% confidence intervals (CI) compared participant cancer rates with population-based, age-, and sex-specific rates as reported by Surveillance, Epidemiology, and End Results (SEER) for the white population during the years 1978–2018 [23]. We were unable to conduct SIR analysis of borderline ovarian tumors because SEER stopped registering these tumors in 2002 [24]. Cox proportional hazards regression models were used to calculate hazard ratios (HR) and 95% CI comparing the exposed with the unexposed; these models used age as the underlying time metric and generally included terms for birth year and original study cohort; when case numbers of exposed or unexposed were <2, analyses were adjusted only for birth year. Follow-up began at birth and continued until the diagnosis of interest or the most recent questionnaire (up to a month, year), whichever occurred first.

Cancer outcomes were assessed overall and by the site of origin. Nonmelanoma skin cancers were omitted from all analyses. Because of inaccurate self-reporting [25,26], cervical cancers were omitted from overall cancer analyses. While recognizing the need for cautious interpretation, females’ self-reports of cervical cancers were assessed in a site-specific analysis due to the important association between prenatal DES exposure and cervical malignancy (CCA). Due to concerns about inaccuracy, mothers’ reports of cervical cancers affecting their daughters were not assessed.

Using data from Third Generation Study participants, we assessed concordance between self-reported cancer diagnoses and mothers’ reports of cancers affecting their daughters. These analyses were confined to the 27 dyads in which the mother or daughter completed a questionnaire after the other had reported the daughter’s cancer diagnosis, and omitted reports of cervical cancer, for which concordance was roughly 30% (due mostly to mothers reporting cervical cancer affecting daughters who self-reported cervical dysplasia).

## 3. Results

The characteristics of Third Generation Study participants, as self-reported on the baseline questionnaire, are shown in Table 1. The majority of participants (84%) were less than 30; more unexposed than exposed females were age 30 or more. More than half of participants were college-educated; the percent reporting a college degree was slightly higher in exposed participants than in the unexposed. Nearly a third of females reported a recent mammogram, and the percent was higher in the unexposed (33.7%) than the exposed (29.8%). Fewer than 10% of females reported having had a breast biopsy; breast biopsy was slightly more common in the unexposed (7.2%) than in the exposed (4.6%). More than 90% of females reported having a pelvic exam, and the percent was similar for the exposed (93.2%) and unexposed (92.6%). Biopsy of the lower genital tract (cervix, vulva, vagina) was slightly more common in the exposed (15.5%) than in the unexposed (13.1%).

### 3.1. Self-Reported Benign Outcomes (Table 2)

Comparing the DES-exposed with the unexposed, the adjusted HR was 1.72 (0.87–3.44) for ovarian cyst (31 exposed, 15 unexposed cases) and 1.29 (0.71–2.35) for PCOS (38 exposed, 21 unexposed cases). The HR for benign ovarian tumors was 1.72 (0.18–16.72) adjusted for birth year (3 exposed, 1 unexposed case). For borderline ovarian tumors, the HR was 3.46 (0.37–32.42) adjusted for birth year (4 exposed, 1 unexposed case). At the time of the ovarian borderline tumor diagnosis, the exposed females were ages 20, 24, 26, and 42; the unexposed female was age 45. The adjusted HR was 2.41 (0.49–11.85) for benign and borderline ovarian tumors combined (7 exposed, 2 unexposed cases).

**Table 2 cancers-16-02575-t002:** Hazard ratios (HR) and 95% confidence intervals (CI) ^‡^ for the relation of DES to benign reproductive tract and breast diagnoses in Third Generation Study participants.

Benign Diagnoses	Exposed N = 964	Unexposed N = 529	HR (95% CI) ^‡^
Ovarian diagnoses			
Ovarian cyst *	31	15	1.72 (0.87–3.44)
PCOS **	38	21	1.29 (0.71–2.35)
Benign ovarian tumor ***	3	1	1.72 (0.18–16.7)
Borderline ovarian tumor	4	1	3.46 (0.37–32.4)
CIN2+ ****	29	21	1.10 (0.58–2.09)
Breast diagnoses			
Atypia ±	4	3	0.89 (0.16–5.03)
Other benign breast tumor ±±	14	11	0.77 (0.32–1.89)
Cyst ±±±	12	4	2.68 (0.74–9.69)

^‡^ HR and 95% CI are adjusted for birth year and study center when cell count for exposed and unexposed is ≥2; otherwise adjusted for birth year only. * Unspecified ovarian cysts. ** HR for polycystic ovarian syndrome is based on 2019 follow-up questionnaire responses (n = 458 exposed, 305 unexposed). *** Benign ovarian tumors include dermoid tumors and lipoleiomyoma. **** CIN2, CIN3, VIN3, HGSIL, and carcinoma in situ are based on pathology reports. ± Breast atypia includes atypical ductal hyperplasia, ductal or lobular carcinoma in situ. ±± Benign breast tumors include fibroadenoma and unspecified benign tumors. ±±± Cysts include fibrocystic breast disease and unspecified cysts.

Of the 29 DES-exposed females with a pathology confirmed diagnosis of CIN2+, 11 had a diagnosis of CIN2, 15 had CIN3, 1 had HGSIL, and 2 had CIS. Of the 21 unexposed females with CIN2+, 7 had a diagnosis of CIN2, 11 had CIN3, 1 had HGSIL, and 2 had CIS. Comparing the exposed with the unexposed, the adjusted HR was 1.10 (0.58–2.09) for confirmed CIN2+. Regardless of DES exposure status, none of the pathology reports indicated moderate/severe dysplasia of the vulva or vagina.

Among third-generation females who self-reported a benign diagnosis based on breast biopsy and compared the exposed to the unexposed, the adjusted HR was 2.68 (0.74–9.69) for breast cyst/fibrocystic disease (12 exposed, 4 unexposed cases). We found no evidence of an elevated risk of benign breast tumors (HR: 0.77; 0.32–1.89) (14 exposed, 11 unexposed cases) or atypical breast diagnoses (HR: 0.89; 0.16–5.03) in relation to DES (4 exposed, 3 unexposed cases).

### 3.2. Self-Reported Malignant Outcomes (Table 3)

The SIR for overall cancer in the DES-exposed females (18 cases) was 1.05 (0.62–1.65) and 1.06 (0.56–1.81) in the unexposed (13 cases). Comparing the DES-exposed with the unexposed, the adjusted HR for overall cancer was 0.83 (0.36–1.90).

**Table 3 cancers-16-02575-t003:** Standardized incidence ratios (SIR), hazard ratios (HR), and 95% confidence intervals (CI) ^‡^ for DES in relation to total cancer and site-specific cancer based on Third Generation Study participants’ self-reported cancer and mothers’ reports of cancer affecting female and male offspring.

Data Source	Exposed Case No.	Exposed SIR (95% CI)	Unexposed Case No.	Unexposed SIR (95% CI)	HR (95% CI) ^‡^
**Self-report (female only) ***					
Total cancer	18	1.05 (0.62–1.65)	13	1.06 (0.56–1.81)	0.83 (0.36–1.90)
Cervical cancer **	2	2.21 (0.27–8.00)	1	1.54 (0.04–8.60)	1.24 (0.11–14.27)
Breast cancer	2	0.53 (0.06–1.90)	6	1.87 (0.69–4.07)	0.12 (0.02–0.65)
Thyroid cancer	6	2.01 (0.74–4.38)	3	1.54 (0.32–4.49)	1.41 (0.30–6.68)
Soft tissue cancer	2	5.49 (0.66–19.83)	0	n/a	n/a
**Mothers’ reports *****					
**Female**					
Total cancer	41	1.12 (0.81–1.52)	36	1.39 (0.97–1.92)	0.80 (0.49–1.32)
Breast	3	0.45 (0.09–1.30)	7	1.16 (0.47–2.38)	0.28 (0.07–1.21)
Thyroid	13	1.96 (1.05–3.36)	5	1.18 (0.38–2.75)	2.13 (0.71–6.42)
Non-Hodgkin lymphoma	4	2.92 (0.80–7.48)	0	n/a	n/a
Ovarian	1	0.90 (0.02–5.01)	0	n/a	n/a
Uterine	1	1.11 (0.03–6.2)	0	n/a	n/a
**Male**					
Total cancer	25	0.79 (0.51–1.16)	14	0.67 (0.37–1.13)	1.41 (0.70–2.86)
Testicular	5	0.91 (0.29–2.11)	4	1.18 (0.32–3.02)	0.83 (0.22–3.21)
Urinary bladder	2	4.57 (0.55–16.52)	0	n/a	n/a
**Combined (Females + Males)**	66	0.97 (0.75–1.23)	50	1.07 (0.79–1.41)	0.97 (0.65–1.45)

^‡^ HR are adjusted for birth year, study center (and sex in analysis combining the sexes). * Self-reported data from 964 exposed and 529 unexposed females participating in the Third Generation study. ** Due to inaccuracy in reporting cervical outcomes, results are limited here to self-report and should be interpreted cautiously. *** Based on mothers’ (second-generation women’s) reports for 5600 exposed (2675 females, 2925 males) and 3042 unexposed (1491 females, 1551 males) third-generation individuals. n/a: Data not available for this analysis.

None of the exposed or unexposed third-generation females reported cancer of the ovary or uterus. The SIR for DES in relation to self-reported cervical cancer was 2.21 (0.27–8.00) in the exposed (2 cases) and 1.54 (0.04–8.60) in the unexposed (1 case). The HR for cervical cancer, comparing the exposed with the unexposed, was 1.24 (0.11–14.27) adjusted for birth year.

The SIR for breast cancer was 0.53 (0.06–1.90) in the DES-exposed (2 cases) and 1.87 (0.69–4.07) in the unexposed (6 cases). Comparing the exposed with the unexposed, the adjusted HR for breast cancer was 0.12 (0.02–0.65).

The SIR for thyroid cancer was 2.01 (0.74–4.38) in the exposed (6 cases) and 1.54 (0.32–4.49) in the unexposed (3 cases); the adjusted HR comparing the exposed with the unexposed was 1.41 (0.30–6.68). Two DES-exposed females reported cancer of the soft tissues (leiomyosarcoma, neurofibrosarcoma); the SIR was 5.49 (0.66–19.83); there were no cases in the unexposed. There was no indication of elevated risks of other malignancies in the exposed. The SIR for leukemia was 0.93 (0.02–5.17) in the exposed (1 case) and 3.13 (0.38–11.29) in the unexposed (2 cases). The HR was 0.29 (0.03–3.32) for DES in relation to leukemia self-reported by the third-generation females.

### 3.3. Malignancies Affecting Third-Generation Females as Reported by Mothers (Table 3)

Based on the mothers’ reports, the SIR for overall cancer was 1.12 (0.81–1.52) in DES-exposed third-generation females and 1.39 (0.97–1.92) in the unexposed (41 exposed, 36 unexposed cases). Comparing the DES-exposed with the unexposed, the adjusted HR for overall cancer risk was 0.80 (0.49–1.32).

Also based on the mothers’ reports, the SIR for breast cancer in the DES-exposed daughters was 0.45 (0.09–1.30) and 1.16 (0.47–2.38) in the unexposed (three exposed, seven unexposed cases). Comparing the DES exposed with the unexposed, the adjusted HR for breast cancer was 0.28 (0.07–1.21). Based on the mothers’ reports, the SIR for thyroid cancer in the DES-exposed daughters was 1.96 (1.05–3.36) and 1.18 (0.38–2.75) in the unexposed; the adjusted HR was 2.13 (0.71–6.42) (13 exposed, 5 unexposed cases). A possible increase of non-Hodgkin’s lymphoma was seen in the exposed daughters (SIR: 2.92; 0.80–7.48) (4 cases), but confidence intervals were wide; there were no cases in the unexposed. The mothers’ reports did not indicate an elevated risk of other cancers, including cancer of the ovary [SIR: 0.90 (0.02–5.01) based on one exposed case] or uterus [SIR 1.11 (0.03–6.2), also based on one exposed case]. The SIR for leukemia was 0.37 (0.01–2.14) in the exposed (1 case) and 4.35 (1.75–8.97) in the unexposed (7 unexposed cases); a direct comparison of exposed with unexposed yielded an HR of 0.09 (0.01–0.76).

### 3.4. Malignancies Affecting Third-Generation Males as Reported by Mothers

Based on the mothers’ reports, the SIR for overall cancer in the third generation was 0.79 (0.51–1.16) in the DES-exposed males and 0.67 (0.37–1.13) in the unexposed males (25 exposed, 14 unexposed cases). Comparing the DES-exposed with the unexposed, the adjusted HR for overall cancer in males was 1.41 (0.70–2.86). The SIR for testicular cancer was 0.91 (0.29–2.11) in the exposed and 1.18 (0.32–3.02) in the unexposed; the HR was 0.83 (0.22–3.21) (5 exposed, 4 unexposed cases). Overall, the age at testicular cancer diagnosis ranged from 17 to 34. The SIR for urinary bladder cancer was elevated in exposed males (SIR: 4.57; 95% CI: 0.55–16.52), but the CI was wide (2 exposed, 0 unexposed cases). None of the mothers reported a son affected by prostate cancer.

### 3.5. Malignancies Affecting Third-Generation Females and Males Combined, as Reported by Their Mothers (Table 3)

In the analyses of overall cancer in third-generation females and males combined, the SIR was 0.97 (0.75–1.23) in the DES-exposed and 1.07 (0.79–1.41) in the unexposed (66 exposed, 50 unexposed cases). Comparing the exposed with the unexposed, the HR for overall cancer was 0.97 (0.65–1.45).

### 3.6. Agreement between Daughters’ Self-Reports of Malignancy and Mothers’ Reports of Malignancies Affecting Daughters

After excluding cervical cancers, we found good agreement between the daughters’ reports of a cancer diagnosis and the mothers’ reports of a cancer affecting the daughter after omitting cervical cancers. Of the 27 dyads in which the mother or daughter completed a questionnaire after the other had reported a daughter’s cancer diagnosis, 21 mother and daughter dyads (78%) were concordant.

## 4. Discussion

In this study, we evaluated the potential intergenerational effects of DES exposure on third-generation offspring, i.e., individuals born to females who were prenatally exposed to DES. Specifically, we assessed cancer risk and the risk of benign tumors of the breast and reproductive tract in third-generation DES-exposed females participating in the NCI Third Generation Study. We also assessed the risk of cancer in third-generation female and male offspring as reported by prenatally DES-exposed and unexposed mothers participating in the NCI DES Follow-up Study. The mothers’ reports expanded our study by providing cancer diagnoses in third-generation males and by identifying cancers in third-generation females who did not participate in the Third Generation Study. We found no evidence of increased risk of overall cancer in third-generation females, regardless of reporting source, and little evidence in males, based on mothers’ reports, but our study is small. Additionally, the third generation is relatively young (most were less than age 50 at the most recent data collection phase), so it has not reached the age of increased overall cancer risk.

Consistent with heritable transmission of DES effects, studies in mice indicate an increased frequency of uterine [13,14] and ovarian [13,15] tumors in the female offspring of perinatally DES-exposed dams [13,15] and sires [14]. Animal studies implicate an array of epigenetic phenomena, including DNA methylation, in the intergenerational transmission of DES effects [16,27,28]. A recent study of postmenopausal females identified a link between prenatal DES exposure and DNA methylation at 10 CpG sites, supporting the possibility that epigenetic alterations persist through life and may mediate DES effects in humans [17]. We cannot, however, entirely exclude the possibility that potential third-generation effects arise from direct DES exposure to primordial germ cells during the mother’s prenatal exposure.

In the current study, we found limited evidence that DES-exposed third-generation females are at increased risk of ovarian cysts, PCOS, or benign ovarian tumors; although the HR for ovarian cysts and benign tumors was elevated, the findings were compatible with chance. In our earlier report, the SIR for borderline ovarian tumors indicated a nearly 15-fold and statistically significant elevated risk based on two self-reported, pathology-confirmed borderline ovarian tumors (0 unexposed cases). In addition, a prenatally DES-exposed female reported a daughter affected by juvenile granulosa cell ovarian cancer at age 7. The present data, based on further follow-up, identified two additional exposed cases of borderline ovarian tumor, for a total of four exposed cases, and one unexposed case (total of one). Although the HR for DES in relation to these tumors was elevated, the CI was wide. Because borderline ovarian tumors are no longer reportable to SEER [24], we were unable to compute SIR based on the updated total case numbers. We note that 3 of the 4 DES-exposed borderline ovarian cases identified to date occurred in young females (ages 20, 22, and 26), whereas a fourth exposed case and the single unexposed case were diagnosed at ages 42 and 45, respectively. Consequently, we cannot entirely dismiss a possible susceptibility to early-onset borderline ovarian tumors in the female offspring of prenatally DES-exposed females. In our study, none of the exposed or unexposed third generation females reported cancer of the ovary, and a study of third-generation females conducted in France found no evidence of an increased risk of ovarian cancer [29]. However, a small cell ovarian cancer in an adolescent female (age 15) has been described in a case report [30].

Females who were exposed in utero to DES are at increased risk of clear cell adenocarcinoma of the vagina/cervix (CCA). The unusual occurrence of this tumor in adolescent and young adult females instigated the study that identified DES as a transplacental carcinogen [4]. None of the DES-exposed third-generation females in the current study reported a CCA diagnosis. However, a recent case report described a CCA tumor in a young female (age 8) whose mother was exposed in utero to DES, suggesting the possibility of intergenerational transmission of CCA risk [31], although direct toxicity to the prenatally exposed mother’s primordial germ cells cannot be ruled out.

Prenatally DES-exposed females are at greater risk of CIN2+ [7,8], but we found little evidence of this association in third-generation females. The percents of DES-exposed and unexposed females reporting pelvic examinations were comparable, suggesting similar screening in the exposed and unexposed females. However, pathology record acquisition was poor overall, with higher retrieval in the unexposed (54%) than in the exposed (47%), which may have introduced bias. We also do not know the extent to which the females in the exposed and unexposed groups were vaccinated against HPV. Given these limitations, our findings with regard to CIN2+ should be considered tentative.

Although the risk of cystic/fibrocystic breast disease appeared to be elevated in DES-exposed third-generation females, the confidence interval was wide. We found no evidence of an increased risk of benign breast tumors or of breast tissue atypia. Similar to a study in France [29], we found no evidence of increased breast cancer risk in the exposed females. Indeed, our data suggested a strong inverse association between DES and breast cancer risk. Somewhat more unexposed than exposed females reported having a mammogram or breast biopsy, which might have contributed in small part to the higher number of breast cancer cases in the unexposed, despite adjustment for birth year. We will continue to monitor breast cancer risk in the cohort as the majority of third-generation females have not reached the age of increased breast cancer risk observed in prenatally DES-exposed women [9,10].

The NCI DES Follow-up Study of prenatally DES-exposed and unexposed females found no evidence of an increased risk of thyroid malignancy in exposed females [6]. Studies of animals have shown changes in thyroid function, although not thyroid malignancy, in the offspring of prenatally DES-exposed females [32]. In the present study, evidence suggesting an association between DES and thyroid cancer in third-generation females was seen primarily in the mothers’ reports. An influence of DES on human endocrine function is plausible and warrants further study.

The mothers’ reports suggested an elevated risk of non-Hodgkin’s lymphoma in exposed daughters, but confidence intervals were wide; no cases were self-reported by the daughters. The daughters’ self-reports suggested an increased risk of soft tissue sarcomas in the exposed, but again, the confidence intervals were wide. We have no explanation for the elevated risk of leukemia in the unexposed observed in the mothers’ and daughters’ reports; this may be a chance finding.

The overall risk of cancer in DES-exposed third-generation males was modestly elevated, but the confidence interval was wide. Prenatally DES-exposed males have an increased risk of testicular cancer, but we found no evidence of this association in the exposed third-generation males. However, most testicular cancers occur between ages 15 and 44 [33], and a minority of third-generation males in our study had reached age 44 at the most recent data collection phase. Additionally, no cases of prostate cancer were reported by the mothers of DES-exposed third-generation males, but given the young age of this cohort, assessment of this cancer is premature. We found no evidence of an increased risk of thyroid cancer in the DES-exposed males. Although the data suggested an increased risk of urinary bladder cancer, the estimate was imprecise.

The health effects of prenatal DES exposure are more extensive in females than in males; consequently, our study focused on the offspring of prenatally exposed females. A strength of our study is that in utero DES exposure or lack of exposure in the mothers of all third-generation individuals was verified by the medical record or a doctor’s note. To the best of our knowledge, the NCI DES study comprises the largest cohorts of individuals with verified DES exposure. A limitation of our study is the unavailability of information about the offspring of prenatally DES-exposed males. Although our conclusions are limited by the small size and young age of the third generation cohort, the importance of monitoring cancer risk in these young individuals is underscored by the strong association between DES and CCA in young DES-exposed second-generation females [5,6,10], and by our previous finding of an elevated risk of borderline ovarian tumors [20]. In this update, we relied on self-report to identify cancer diagnoses, which may have introduced outcome misclassification; registry linkages are in progress and will address this potential limitation in the future. We also relied on self-report for most benign conditions, and the retrieval of pathology records for the CIN2+ analysis was suboptimal. Although we cannot assess mother-daughter concordance for benign diagnoses, mother-daughter concordance regarding cancer diagnoses (excluding cervical cancers) was good, offering some assurance of the validity of our findings.

In summary, our study provides little evidence of increased overall cancer risk in third-generation females or males. Although we found limited evidence of an association with cervical disease, cautious interpretation is warranted due to suboptimal retrieval of pathology reports confirming CIN2+. None of the third-generation females were affected by CCA, but our sample size is small for assessing this rare cancer. We found only limited support for an increased risk of borderline ovarian tumors in the exposed, but the occurrence of these tumors in young females is concerning, given the early onset of CCA diagnoses in prenatally exposed females. DES exposure was inversely associated with breast cancer risk, but elevated risk in the second generation was limited to older women; consequently, third-generation females are too young for a meaningful assessment of this outcome. Based on the mothers’ reports, we found limited evidence of an increased risk of overall cancer in the DES-exposed males. There was no evidence of an elevated risk of testicular cancer in the DES-exposed third-generation males, and none of the exposed mothers reported prostate cancer in a son. However, the majority of third-generation males have not passed through the high-risk ages for testicular cancer and are too young for a meaningful assessment of prostate cancer.

Mature evidence of the relationship between DES exposure and third-generation health outcomes such as cancer requires further follow-up of this potentially vulnerable cohort. Future studies of this and subsequent generations of DES-exposed individuals may have implications for the effects of other pharmaceutical or environmental endocrine disruptors and provide a resource for investigating the mechanisms of intergenerational inheritance.

## 5. Conclusions

To date, there is little evidence that DES is associated with cancer risk in the offspring of prenatally exposed mothers. However, these individuals are relatively young, and long-term monitoring of health outcomes is needed as they age into a period of increased cancer risk.

## Figures and Tables

**Table 1 cancers-16-02575-t001:** Third Generation Study females’ self-reported characteristics * at combined baseline, according to the mothers’ prenatal DES exposure status.

	Mother’s Prenatal DES Exposure Status	
Characteristic	Exposed (n = 964)	Unexposed (n = 529)
	No. (%)	No. (%)
Age		
<20	132 (13.7)	72 (13.6)
20–24	413 (42.8)	202 (38.2)
25–29	268 (27.8)	146 (27.6)
30–34	86 (8.9)	66 (12.5)
≥35	65 (6.7)	43 (8.1)
Education **		
HS graduate or less	345 (36.0)	220 (41.6)
Post HS	60 (6.3)	29 (5.5)
College graduate	553 (57.7)	280 (52.9)
Mammogram ***		
No	657 (70.2)	339 (66.3)
Yes	279 (29.8)	172 (33.7)
Breast biopsy ****		
No	918 (95.4)	491 (92.8)
Yes	44 (4.6)	38 (7.2)
Pelvic exam ****		
No	64 (6.8)	38 (7.4)
Yes	875 (93.2)	477 (92.6)
Lower genital tract biopsy ****		
No	759 (84.5)	433 (86.9)
Yes	139 (15.5)	65 (13.1)

* Texas did not participate in the 2019 baseline data collection. ** Abbreviation: HS, high school. *** Mammogram asked on baseline questionnaires only; the question specified “in past 5 years”. **** Asked on all questionnaires; frequencies based on the most recent questionnaire response. Baseline questionnaires asked “ever”; 2009 follow-up questionnaires specified past 10 years; 2019 follow-up questionnaires specified since last questionnaire (date of last questionnaire was shown on the questionnaire).

## Data Availability

The original data presented in the study are openly available in the Division of Cancer Epidemiology and Genetics at the National Cancer Institute’s Publication Data Repository URL.

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
