# Peer review of "Benign and Malignant Outcomes in the Offspring of Females Exposed In Utero to Diethylstilbestrol (DES): An Update from the NCI Third Generation Study"

_cancers, 2024, doi:10.3390/cancers16142575_

Round 1

Reviewer 1 Report

Comments and Suggestions for Authors

Cancers (Manuscript ID: cancers-3088868), Comments to the Authors:

Title: Benign and malignant outcomes in the offspring of females exposed in utero to diethylstilbestrol (DES). An update from the NCI Third Generation Study.

Comments

The submitted manuscript focused on investigating females exposed prenatally to diethylstilbestrol (DES) who have elevated risk of cervical dysplasia, breast cancer, and clear cell adenocarcinoma (CCA) of the cervix/vagina. Testicular cancer risk is increased in prenatally exposed males. Epigenetic changes may mediate transmission of DES effects to the next (“third”) generation offspring. The authors used data self-reported by third generation females, and assessed DES in relation to risk of cancer and benign breast and reproductive tract conditions. Using data from prenatally DES-exposed and unexposed mothers, the authors assessed DES in relation to cancer risk in their female and male offspring. Cancer risk was assessed by standardized incidence ratios (SIR) and 95% confidence intervals (CI); risks of benign and malignant diagnoses were assessed by hazard ratios (HR) and 95% CI. DES exposure did not increase risk of overall cancer (HR 0.83; CI 0.36-1.90), breast cancer, or severe cervical dysplasia. No females reported CCA. Risk of borderline ovarian cancer appeared elevated, but the HR was imprecise (3.46; CI 0.37-32.42). Based on mothers’ reports, DES exposure did not increase risk of overall cancer (HR 0.80; CI 0.49-1.32) or of other cancers in third generation females. Overall cancer risk in exposed males appeared elevated (HR 1.41; CI 0.70-2.86) but the CI was wide. Risk of testicular cancer was not elevated in exposed males; no males were affected by prostate cancer.

I think the submitted manuscript can be accepted for publication after the authors responds to the following comments:

1.     The authors mentioned that “the third generation are relatively young”, so why did initiate the investigation if they know that probability of finding cancer in such young population is low.

2.     The authors should include a section “limitation of the study” and they should include all limitations of their study.

3.     The authors should include the future perspectives of tehri work and recommendations.

4.     Why did the authors have 2 conclusions in their submitted paper.

5.     The authors should include figures summarizing their findings.   

Author Response

Reviewer 1

Comment 1: “The authors mentioned that “the third generation are relatively young”, so why did initiate the investigation if they know that probability of finding cancer in such young population is low.”

Response 1: The importance of investigating cancer risk in this young population is linked to two key facts:  1) the emergence of vaginal clear cell adenocarcinoma in prenatally DES exposed girls and young women (age 9+); and 2) the diagnosis of juvenile granulose and borderline ovarian cancer in the third generation women (age 7+).  We appreciate the reviewer alerting us to this issue, and revised this area (see track changes, paragraph 1, page 23).

Comment 2: “The authors should include a section “limitation of the study” and they should include all limitations of their study.”

Response 2: The limitations of our study are described in context with related discussion points; for example, second sentence at the top of page 18 (“…but our study is small.  Additionally, the third generation is relatively young (most were less than age 50 at the most recent data collection phase), so has not reached the age of increased overall cancer risk.”), in the second full paragraph on page 20 (“…which may have introduced bias. We also do not know the extent to which the females in the exposed and unexposed groups were vaccinated against HPV.  Given these limitations, our findings with regard to CIN2+ should be considered tentative.”), and the first full paragraph on page 21 (“…but given the young age of this cohort, assessment of this cancer is premature,”) and are noted in the summary paragraph on pages 22-23 (especially see track changes). 

Comment 3: “The authors should include the future perspectives of their work and recommendations.”

Response 3: We appreciate the reviewer’s suggestion. We added the following sentences to the next-to-last paragraph on page 24 (see track changes):  

“Future studies of this and subsequent generations of DES-exposed individuals may have implications for the effects of other pharmaceutical or environmental endocrine disruptors, and provide a resource for investigating the mechanisms of intergenerational inheritance.”  We also added the need for long term follow-up of this cohort as they age (see track changes in Conclusions on page 24).

Comment 4: “Why did the authors have 2 conclusions in their submitted paper.”

Response 4: We wrote an overall conclusion at the end of our Discussion section, which was followed by the formal Conclusion section required by the journal. We thank the reviewer for noting this awkward juxtaposition and revised our final paragraph in the Discussion section to begin with the words “In summary” rather than “In conclusion.”

Comment 5: “The authors should include figures summarizing their findings.”

Response 5: We agree that figures are useful for visually summarizing and comparing complex and diverse findings.  Because our results are straightforward and many findings are null, we feel that figures would not provide additional information.  However, we appreciate the reviewer’s request for additional data displays and have expanded Table 3 to include the site-specific cancers of interest. 

Reviewer 2 Report

Comments and Suggestions for Authors

The manuscript titled "Benign and malignant outcomes in the offspring of females exposed in utero to diethylstilbestrol (DES). An update from the NCI Third Generation Study" provides valuable insights into the long-term health impacts of DES exposure across generations. The study is comprehensive and adds significant knowledge to the existing literature on intergenerational effects of endocrine disruptors. However, there are several areas that need improvement to enhance the clarity and impact of the manuscript.

1. The results section is extensive and detailed. However, it would greatly benefit from the inclusion of figures to visually represent key findings. Graphs and charts could illustrate the comparative risks of cancer and benign conditions between DES-exposed and unexposed groups, making it easier for readers to grasp the significant points at a glance.

2. The number of references cited is relatively low given the scope and depth of the study. Increasing the number of references will strengthen the manuscript by providing additional context and supporting evidence from other relevant studies. It may also be beneficial to cite some of my previous work on related topics to provide a broader perspective and underscore the study's relevance within the existing body of research.

3. The introduction is well-written and provides a clear background on DES exposure. However, it could be improved by briefly discussing more recent studies on intergenerational effects of endocrine disruptors.

4. The conclusion is concise but should emphasize the need for ongoing monitoring of the cohort as they age. This will help in understanding the long-term impacts more comprehensively.

5. Please consider to citing this work doi: 10.1007/s00432-023-04863-3.

Author Response

Reviewer 2

Comment 1: “The results section is extensive and detailed. However, it would greatly benefit from the inclusion of figures to visually represent key findings. Graphs and charts could illustrate the comparative risks of cancer and benign conditions between DES-exposed and unexposed groups, making it easier for readers to grasp the significant points at a glance.”

Response 1: As noted above, figures are useful to visually summarize and compare complex or diverse findings.  However, our results are straightforward, many findings are null, and in our judgment, figures would not provide additional information.  We agree though with the reviewer’s suggestion that the information in the Results would benefit by being presented in the tables. Thus, we expanded Table 3 to include the site-specific cancer findings.

Comment 2: “The number of references cited is relatively low given the scope and depth of the study. Increasing the number of references will strengthen the manuscript by providing additional context and supporting evidence from other relevant studies. It may also be beneficial to cite some of my previous work on related topics to provide a broader perspective and underscore the study's relevance within the existing body of research.”

Response 2: Reviewer 2 suggested citing one of their publications.  The suggested citation reports a specific laboratory finding; which, in our judgment, is too esoteric for this manuscript.  However, we appreciate the reviewer’s suggestion of citing literature to support ongoing studies in the field of epigenetics and intergenerational or transgenerational transmission.  Consequently we have additionally cited two publications that more broadly review the broad scope of laboratory and human studies on epigenetics and intergenerational or transgenerational inheritance (Nicolella et al., and Nilsson et al; new references 18 and 19).  

Unfortunately, little research has been published on benign and malignant tumors in the third generation of DES-exposed individuals, limiting our ability to cite directly relevant literature.  However, we have added a relevant article by Tournaire et al. (new reference 29), which we had inadvertently omitted.

Comment 3: “The introduction is well-written and provides a clear background on DES exposure. However, it could be improved by briefly discussing more recent studies on intergenerational effects of endocrine disruptors.”

Response 3: The absence of more recent animal studies supporting epigenetic inheritance with DES exposure is an ongoing challenge for epidemiologists studying intergenerational DES effects.  However, as noted above, we have additionally cited a review by Nicollela et al. describing animal and human studies supporting intergenerational health effects of other environmental endocrine disruptors (page 5), as well as an article by Nilsson et al. summarizing environmental factors and transgenerational inheritance (see track changes on page 6, and new references 18-19). 

Comment 4: “The conclusion is concise but should emphasize the need for ongoing monitoring of the cohort as they age. This will help in understanding the long-term impacts more comprehensively.”

Response 4: We appreciate and agree with the reviewer’s suggestion; the importance of further follow-up is now emphasized in the first full paragraph on page 24 (see track changes) and in the Conclusions on page 24 (see track changes).  

Round 2

Reviewer 1 Report

Comments and Suggestions for Authors

Cancers (Manuscript ID: cancers-3088868 - Revised Version), Comments to the Authors:

Title: Benign and malignant outcomes in the offspring of females exposed in utero to diethylstilbestrol (DES). An update from the NCI Third Generation Study.

Comments

After reading the authors response to my comments, I think the revised version of the submitted manuscript can be accepted for publication.

Reviewer 2 Report

Comments and Suggestions for Authors

The paper has been significantly improved.